# Synthesis of Protein-Oligonucleotide Conjugates

**DOI:** 10.3390/biom12101523

**Published:** 2022-10-20

**Authors:** Emma E. Watson, Nicolas Winssinger

**Affiliations:** 1Department of Chemistry, School of Physical Sciences, The University of Adelaide, Adelaide, SA 5005, Australia; 2Department of Organic Chemistry, Faculty of Science, NCCR Chemical Biology, CH-1205 Geneva, Switzerland

**Keywords:** oligonucleotides, proteins, bioconjugation

## Abstract

Nucleic acids and proteins form two of the key classes of functional biomolecules. Through the ability to access specific protein-oligonucleotide conjugates, a broader range of functional molecules becomes accessible which leverages both the programmability and recognition potential of nucleic acids and the structural, chemical and functional diversity of proteins. Herein, we summarize the available conjugation strategies to access such chimeric molecules and highlight some key case study examples within the field to showcase the power and utility of such technology.

## 1. Introduction

Nucleic acids and proteins form the key classes of functional molecules in biology and thus, it stands to reason that each of these biomolecules serves as an important building blocks for biological chemistry and chemical biology. However, just like in biology, the classes are often exploited to different ends, based on their fundamentally different properties (Figure 1). The limited number of structural components which comprise nucleic acids, in concert with their defined hydrogen bonding properties, afford them unique programmability and ease with which to predict overall geometry [1]. For these reasons nucleic acids are often used for specific recognition of partner strands enabling the formation of defined nano objects, or for the physical transfer of information through conformational change by acting as molecular circuitry. It is important to note that such systems can rely equally on native DNA or on designer nucleic acids (such as peptide nucleic acids (PNA) [2,3,4,5,6,7,8,9,10] or threoninol nucleic acids [11]) to fulfill this role. In contrast, proteins have a much wider array of potentially accessible structures as a result of their larger number of component building blocks, as well as the broader range of interactions these can undergo to influence global structure [1]. However, this broader molecular space available to proteins makes them inherently more difficult to program for specific shapes and interactions (although significant advances are being made in this area [12,13,14,15]). By creating hybrid structures, composed of both nucleic acids and proteins it is possible to harness the power of each to create uniquely powerful systems.

## 2. Strategies for Conjugation

While nucleic acids and proteins have tremendous power and utility from a functional perspective, this can be infinitely expanded through their combination. [1,16,17,18,19] However, this is predicated upon the ability to undertake such combination in a programmable and controllable manner. As such, the fundamental conjugation chemistries available are the ultimate determining factors for the success of such an endeavor. Undertaking precision chemistry on biological molecules is inherently challenging, given the ubiquity of many functional groups and thus only a specific range of chemical modifications are applicable in this setting. Broadly, these can be classified as either purely chemical transformations or biochemical transformations (i.e., those assisted by the presence of an enzyme or by supramolecular interactions such as streptavidin-biotin [20]) (Figure 2). Below we discuss the most widely exploited means of protein-oligonucleotide conjugation. While these transformations are depicted with a particular directionality (i.e., which reactive partner is present on which biomolecule) it should be noted that in many cases (particularly for the chemical conjugation methods), this connectivity can be reversed, thereby broadening the potential options for conjugation.

### 2.1. Chemical Conjugation Strategies

While the vast majority of strategies for conjugation of proteins and nucleic acids are chemical in nature, the subset of these reactions applicable in a given setting relies upon the choice of whether to exploit a naturally occurring reactive handle (predominantly lysine or cysteine) or an unnatural one. While naturally occurring moieties present practical ease in terms of substrate access, they can suffer from difficulties in specificity. Conversely access to unnaturally occurring reactive handles provides chemical specificity but at the expense of challenges in obtaining the starting protein.

The most widely utilized strategy for targeting naturally occurring reactive handles is conjugation through the nucleophilic amine side chain of lysine residues (Figure 2A). This is predominantly achieved through the functionalization of the oligonucleotide with an N-hydroxysuccinimide ester which following nucleophilic attack yields a stable amide linkage [8,21,22,23,24,25,26,27]. Cysteine residues are also commonly used as nucleophilic handles for protein modification with maleimide functionalized oligonucleotides (Figure 2B) [2,24,25,26,28,29,30,31,32]. While this is a commonly exploited strategy, through the conjugate addition of the cysteine thiol to yield to a thioether linkage, this type of conjugation is inherently unstable due to the acidity of the α-proton, which can lead to the elimination of the thiol residue and decomposition of the conjugate. Cysteine can also participate in the linking of protein and oligonucleotides through the formation of disulfide bonds (Figure 2C) [33,34]. Such conjugation is often assisted through the initial formation of a reactive disulfide intermediate (such as pyridylthiol) to drive the formation of the desired stable, disulfide conjugate. While disulfides are easily accessible from a synthetic standpoint, their reversibility must also be considered, especially in the harsh redox environment of a biological system. Finally, N-terminal cysteine residues undergo native chemical ligation (NCL) in the presence of an oligonucleotide bearing a C-terminal thioester (Figure 2D) [3,4,35]. An initial trans-thioesterification step links the two components, which is followed by a rapid S-to-N acyl shift to generate the stable amide bond linkage and regenerate the cysteine side chain.

The incorporation of non-native functional handles can either be achieved through bifunctional linker chemistry or through the incorporation of unnatural amino acids (UAAs). Either of these strategies can be used to incorporate azide reactive handles into protein sequences, including via the UAA pAzF (p-azidophenylalanine) [11,36]. This can then subsequently be exploited for copper-catalyzed azide-alkyne cycloaddition (CuAAC) [37] with an alkyne functionalized oligonucleotide or strain-promoted azide-alkyne cycloaddition (SPAAC) [11,24,38,39] by exploiting a strained cyclooctyne (Figure 2E). Another commonly exploited UAA is p-acetylphenylalanine (pAcF) [7,31] which facilitates site specific incorporation of a ketone into a protein which can then undergo oxime ligation chemistry [32,40] with an alkoxyamine functionalized oligonucleotide (Figure 2F). Finally, the incorporation of a tetrazine derivatized phenylalanine residue facilitates conjugation to a trans-cyclooctene functionalized oligonucleotide via an inverse electron demand Diels Alder reaction (IEDDA) (Figure 2G) [24].

### 2.2. Biochemical Conjugation Strategies

There are also a wide variety of biochemical-based conjugation strategies available for the construction of protein-oligonucleotide conjugates. One of the most widely applied strategies of this type utilizes enzymes which recognize a small molecular tag and transfer it to a specific peptide sequence within a larger protein substrate (Figure 2H). These enzymes only consider the molecular tag but are promiscuous relative to the nature of the appended unit. Thus, oligonucleotides functionalized with the appropriate tag can be conjugated to the target protein. One of the most well-known enzymes of this type is the Sortase A [10,41,42,43] enzyme which recognizes a LPXTG tag on the C-terminus of a target protein and conjugates it to molecules bearing an N-terminal poly-glycine tag. Similarly, the asparaginyl endopeptidase OaAEP1 can transfer a molecule bearing an N-terminal GL tag to a protein containing a NGL consensus sequence through the liberation of the C-terminal GL dipeptide. [9] Enzymes can also be used to create a reactive intermediate such as in the case of the formylglycine generating enzyme (FGE) which recognizes a CXPXR consensus sequence and converts the requisite cysteine residue into a formylglycine, allowing for the site-specific placement of a reactive aldehyde within the protein sequence. [40] This residue can then react with an incoming nucleophile such as an alkoxyamine to facilitate conjugation. Other classes of enzymes used for protein oligonucleotide conjugation rely on the transfer of post-translational modification. For example, a protein farnesyl transferase (PFTase) can transfer an oligonucleotide labelled with farnesyl pyrophosphate to the cysteine residue of a CVIA consensus sequence [37], while Hedgehog autoprocessing domains can transfer a steroid labelled oligonucleotide to a fused protein of interest, followed by self-cleavage [44,45].

Finally, biochemical conjugation of proteins and oligonucleotides can be accessed via self labelling proteins (Figure 2I). These recognize a small molecule tag and (in contrast to the small tag category) transfer this label onto themselves. The two most common enzymes of this class are the HaloTag [46,47] (which recognizes a haloalkane ligand) and the SnapTag [5,6,46,48] (which recognizes a modified O^6^-benzylguanine nucleobase). Both enzymes are sufficiently promiscuous beyond the recognition of the tag to facilitate the conjugation of tags bearing large substituents, including oligonucleotides.

### 2.3. Hybrid Conjugation Strategies

While each of the individual conjugation strategies highlighted above has the potential to efficiently achieve conjugation between protein and nucleic partners, these individual strategies can often be used to greater effect when combined. The simplest way this can be achieved is through the use of bifunctional linkers, harnessing a pair of orthogonal chemical transformation [26,31,34,49]. Conversely, individual chemical conjugation techniques can be coupled with a templating element (ligand, protein–protein interaction, antibody, metal chelation) [3,4,22,27,32,50,51,52,53] which can be used to direct the conjugation to a specific region on a target protein, thereby improving the selectivity of the process (Figure 3). This strategy relies on the recognition element (red) which binds to the target and displays an oligonucleotide strand. The complimentary conjugation oligonucleotide strand (blue) then hybridizes with the first strand, bringing it close to the desired region on the protein surface in order to undergo covalent conjugation, followed by dissociation of the templating moiety. While conventional NHS chemistry can be exploited in this setting [50], most commonly photocrosslinking motifs (such as diazirines [51], aryl azides [52,53] and benzophenones [49]) are used to enable regiochemical specific conjugation without specific requirements for chemical reactivity.

## 3. Applications of Protein-Oligonucleotide Conjugates

With such a wealth of conjugation strategies available it thus becomes possible to fully exploit the potential of oligonucleotide protein conjugates for a vast array of applications. One of the broadest classes of applications is to enable the specific 1:1 functionalization of proteins [27,50] (particularly antibodies) with functional handles including fluorophores, [51,53] pulldown handles, [52] radiolabels [8,10,42] and nucleic acid barcodes [4]. Fluorescently labelled proteins are particularly applicable and can be leveraged for downstream applications such as sensors [29] and for the transient cell surface labelling required for the high-resolution imaging such as DNA-paint [3,20]. Protein-oligonucleotide conjugates can also be applied to the sensing of nucleic acids through acting as primers for PCR-based detection systems [21,31,48]. In addition to these readout applications protein-oligonucleotide conjugates can also be utilized for structural purposes. These predominantly involve the site-specific localization of target proteins to structures including DNA origami, [2,6,9,11,23,24,37,46,49] arrays, [35] nanoparticles [25] or even the surfaces of phages [41] to impart new activity. The oligonucleotide tags can also be used to bring labelled protein units together to form new entities with precise three-dimensional arrangements, [7,39] such as the characteristic Y-shape to be able to mimic antibody binding [26,40] or to recapitulate enzyme activity [22,28,33,45]. Herein we select several examples as case studies to highlight the utility of these biomolecular systems.

### 3.1. Case Study 1

While inherent challenges exist in terms of the site specificity available through purely chemical conjugation approaches these can be overcome through careful system design. This is elegantly showcased in the work of Ricci and Gothelf, [38] whereby they are specifically able to target the hinge region of IgG1 antibodies for oligonucleotide conjugation. This is achieved through the use of a Lysine Directed Labelling Reagent (LDLR) previously developed by Gothelf [54] which is predisposed towards reactivity at the lysine-rich hinge region of IgG1. The reagent is based upon a salicaldehyde moiety (Figure 4) which undergoes reversible covalent interaction with surface exposed lysine residues to form a corresponding iminium ion. Crucially, this iminium ion then causes activation of the adjacent *p*-phenol ester, increasing its reactivity towards nucleophilic attack by nearby lysine residues. Without the initial iminium ion formation, such reactivity is not sufficient and thus conjugation is only achieved when two lysine residues are found in close proximity, such as in the hinge region of IgG1. Through functionalization of the LDLR with an azide and subsequent SPAAC conjugation to a DBCO (dibenzocyclooctyne) labelled oligonucleotide, the authors where able to achieve site selective conjugation on IgG1 antibodies. They then exploited these chimeric molecules to facilitate programmable protein template reactions by choosing pairs of IgG1 antibodies specific to orthogonal regions of the same target protein. The appended oligonucleotides could then be used to recruit a corresponding pair of reactant strands, bearing the chemical reaction partners. Specifically, the authors used an azide and alkyne pair, which when brought into to close proximity by the oligonucleotide scaffold underwent CuAAC to conjugate the two reactant strands.

### 3.2. Case Study 2

While it is possible to achieve protein-oligonucleotide conjugation in a specific manner harnessing only chemical reactivity, it is a far more common and general approach to combine this with a site specific non-covalent interaction. This initial templation step thereby provides the regiospecificity for the chemical transformation. A powerful example of this strategy is demonstrated by de Greef and coworkers [49] to enable the display of any antibody or Fc fusion protein on DNA origami architectures. This platform is underpinned by the specificity of protein G for the Fc domain of antibodies (Figure 5). The authors engineered the immunoglobulin binding protein to contain an unnatural *p*-benzoylphenylalanine (BPA) in the Fc binding site, which upon UV irradiation enables site-specific covalent conjugation as originally reported by Tsourkas and coworkers [55]. Protein G was also engineered to contain a single N-terminal cysteine residue, allowing for conjugation to an amino functionalized oligonucleotide through a heterobifunctional sulfosuccinimidyl 4-(*N*-maleimidomethyl) cyclohexane-1-carboxylate (Sulfo-SMCC) linker. The authors thereby use Protein G as a small and easily engineerable adaptor protein to enable specific conjugation of oligonucleotides to both antibodies, as well as other proteins of interest expressed as fusion proteins with the Fc domain. They exploit this to enable display of these proteins on the surface of DNA origami structures through complementary base pairing with the conjugate nucleic acid tag.

### 3.3. Case Study 3

As shown above, protein–protein interactions provide a powerful means of templation to enable site-specific chemical conjugation between oligonucleotides and proteins. However, these interactions are not limited to large protein structures, with precisely designed isolated secondary structural elements capable of affording similar templation. This is demonstrated by Seitz and coworkers [4] in their use of coiled-coil protein interactions for the labelling of cell surface membrane proteins (Figure 6). The authors exploit the coiled-coil tag system pioneered by Matsuki [56] to genetically encode orthogonal coiled-coil acceptor units (blue) on the N-terminus of two cancer associated receptors, ErbB2 and EGFR. The corresponding coiled-coil donor sequences were then accessed synthetically as an mercaptophenylacetic acid (MPAA) thioester. These were linked to the oligonucleotide portion (the synthetic oligonucleotide mimic peptide nucleic acid (PNA)) via SPAAC. Upon addition of the functionalized donor strand to the expressed acceptor strand, the two strands are brought into close proximity by the specific coiled-coil interaction. This places the N-terminal cysteine residue from the acceptor strand nearby the synthetic thioester containing strand, leading to proximity templated native chemical ligation (NCL). This proceeds through an initial trans-thioesterification step which transfer the PNA to the acceptor strand and liberates the donor strand. This is then followed by a rapid intramolecular *S*-to-*N* acyl shift to generate a stable amide bond, conjugating the PNA to the cell surface receptor. The authors exploited this system for the erasable labelling of these cancer-associated receptors through the use of an erasable imaging complex in which the fluorescent label is present on a strand which is bridged by a third strand. The binding of the labelled strand can then be erased by the addition of a fourth strand due to the presence of an overhang on the labelled strand.

### 3.4. Case Study 4

Enzymes can also provide a powerful means of conjugating oligonucleotides to proteins. Callahan and coworkers [45] provide an elegant demonstration of this through their exploitation of the Hedgehog autoprocessing domain (HhC) (Figure 7). HhC natively catalyses two linked activities: (1) the cleavage of an N-terminal protein and (2) the ligation of a sterol to the C-terminal residue of the departing protein fragment. The authors were able to hijack this reactivity to allow for the C-terminal functionalization of proteins with so called “steramers” (sterylated DNA), thereby providing an enzymatic means of conjugating nucleic acids to proteins. The authors exploit this system in a subsequent publication [44] for the detection of mutations within the spike protein of SARS-CoV2. This is facilitated through the production of an enzymatic beacon (E-beacon), through the conjugation of a conventional molecular beacon architecture bearing a quencher moiety (Q), to the ATP-independent bioluminescent nanoluciferase Nluc. In the closed state of the beacon the bioluminescence produced by Nluc upon turnover of the furimazine substrate is quenched, and therefore no bioluminescence is observed. However, in the presence of the target nucleic acid strand the molecular beacon exists in the open state, thereby removing the quencher from the vicinity of Nluc, thereby turning on the bioluminescence, and providing a highly sensitive nucleic acid detection platform.

### 3.5. Case Study 5

Luciferase enzymes not only provide the potential for use in detection-based systems but can also be used to initiate photochemical transformations, hence providing a common and attractive target for oligonucleotide conjugation. Winssinger and coworkers [5] demonstrated this through the development of the luciferase-based photocatalysis induced via nucleic acid template (LUPIN) system (Figure 8). This system is underpinned by the bioluminescence resonance energy transfer (BRET) from the Nluc donor to a ruthenium photocatalyst, which then undergoes photoexcitation to facilitate reductive cleavage of a pyridinium linker to liberate either a drug or fluorophore. In order to enable efficient energy transfer from the Nluc to the photocatalyst, ensuring the correct geometry of the system is vital. This is achieved by the precise positioning of multiple protein and PNA components. Firstly, the Nluc is expressed as a fusion protein with dihydrofolate reductase (DHFR) and the SNAP protein, which forms the geometric foundation of the system. The second key component is a synthetic linker bearing an O^6^-benzylguanine residue on one end to facilitate SNAP labelling and a methotrexate ligand on the other end to bind to DHFR, thereby locking the geometry of the system. Within the linker assembly the photocatalyst itself is immobilized in close proximity to the Nluc BRET donor as well as a template PNA strand. This then recruits a complementary substrate PNA strand, positioning the functional warhead and its pyridinium linker in close proximity to the photocatalyst to enable efficient cleavage. While this system has several crucial components, its effectiveness is underpinned by the precise three-dimensional arrangement of its components, which is enabled by the dual conjugation of the PNA linker strand, both by SNAP and ligand binding. The requirement for this precise positioning makes the system responsive to ligand binding, and the presence of an analyte (methotrexate) compete for binding to cpDHFR and turns off the catalysis.

## 4. Discussion

Proteins and oligonucleotides both serve as important functional molecules in biological environments, facilitating programmable functionality. By harnessing the complementary properties of these two classes of biomolecules it is possible to create bespoke systems capable of a vast range of functional outputs. However, underlying each of these systems is the fundamental ability to join the building block biomolecules together in a specific and controlled manner. It is thus the conjugation chemistry which underpins each of these systems. As such the continued growth in the area of bioconjugation is fundamental to the continued expansion of hybrid protein-oligonucleotide systems, particularly in terms of providing site specific conjugation without the need for genetic reengineering. Given the prevalence of photocatalysis in other areas of bioconjugation it is anticipated that such reactivity will enjoy greater application for the synthesis of protein-oligonucleotide conjugates over the coming years, potentially facilitating the desired site selectivity without the difficulties associated with accessing the substrate. Improved conjugation methodologies may also lead the field to shift away from the non-specific and reversible reactivities which have long formed a mainstay in the area, although complete retirement of these techniques is unlikely. Overall, in the areas of the synthetic biomolecules there is an increasing trend towards increased chemical rigor in modification and characterization which can only be of benefit towards accessing high-quality conjugated biomolecules for downstream applications. As we begin to view biomolecules through the lens of chemical building blocks this can only inspire more creativity within the area of protein-oligonucleotide conjugates and allow the community to build ever more complex and elegant systems for functional exploitation.

## Figures and Tables

**Figure 1 biomolecules-12-01523-f001:**
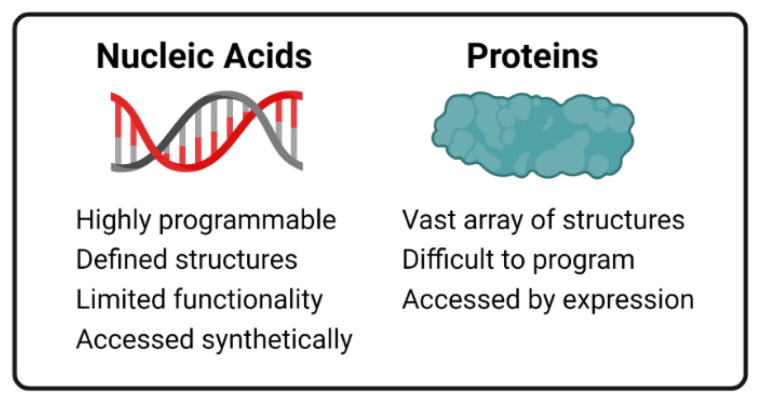
Comparison of nucleic acids and proteins as molecular building blocks.

**Figure 2 biomolecules-12-01523-f002:**
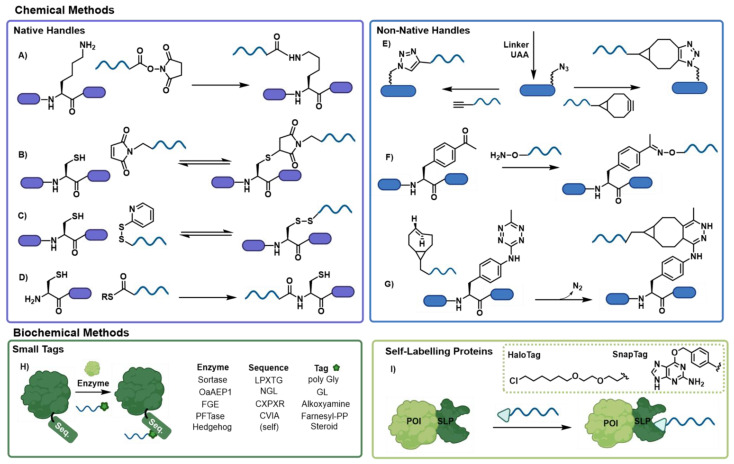
Conjugation chemistries utilized in the formation of protein oligonucleotide conjugates including (**A**) NHS-esters, (**B**) maleimides, (**C**) disulfide bond formation, (**D**) NCL, (**E**) CuAAC and SPAAC, (**F**) oxime ligation, (**G**) tetrazine click, (**H**) enzyme conjugation and (**I**) self-labelling proteins.

**Figure 3 biomolecules-12-01523-f003:**
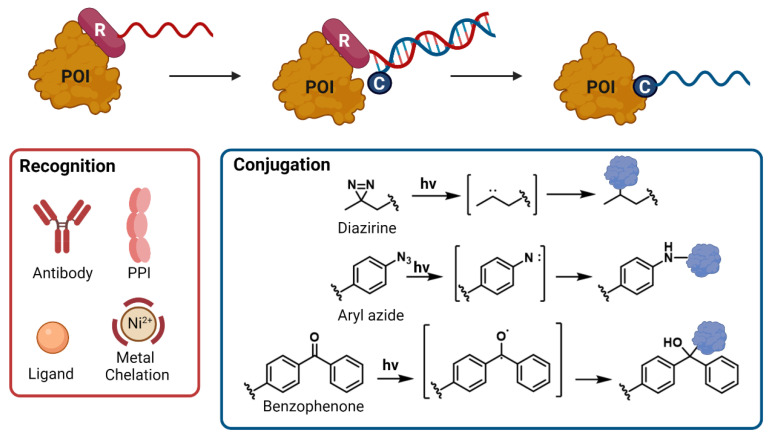
Templated conjugation relies on the regiospecific binding of a recognition motif to the protein surface, thereby imparting regiospecificity to the conjugation event. R: see red box for examples of interaction; C: see blue box for examples of crosslinking moieties; POI: protein of interest; PPI: protein-protein interaction.

**Figure 4 biomolecules-12-01523-f004:**
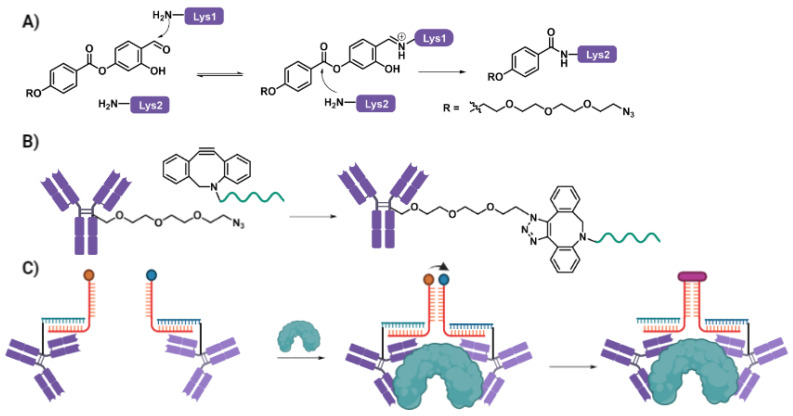
Case Study 1: lysine directed labelling of IgG1 for templated reactivity. (**A**) Initial iminium ion formation activates the p-phenyl ester towards attack from a second nearby lysine residue, enabling site specific conjugation to the hinge region of IgG1 antibodies (**B**). This can be exploited for proximity templated chemistry through the use of an orthogonal pair of antibodies against a single target (**C**).

**Figure 5 biomolecules-12-01523-f005:**
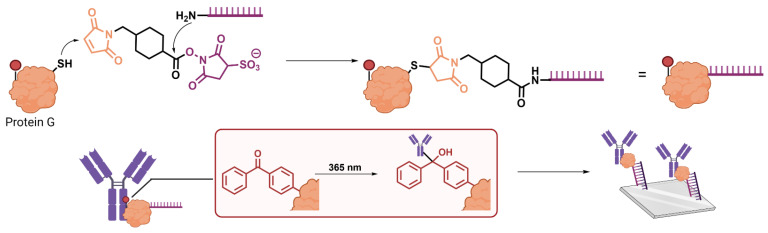
Case Study 2: engineered Protein G-oligonucleotide conjugates as generic handles for labelling antibodies and Fc-fusion proteins. A bifunctional linker is used to conjugate oligonucleotides to a mutant Protein G containing a single cysteine residue. The specificity of Protein G for the Fc domain of antibodies is then harnessed to facilitate site specific conjugation through a BPA residue (represented as a red circle).

**Figure 6 biomolecules-12-01523-f006:**
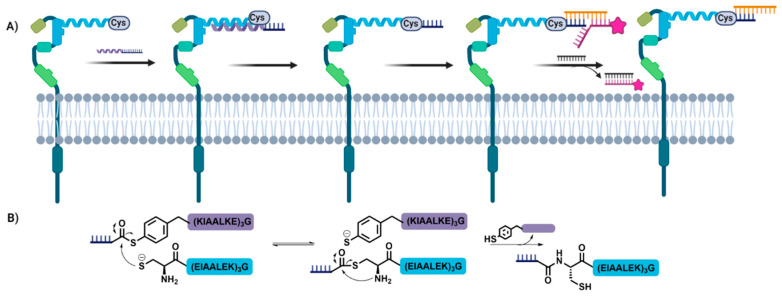
Case Study 3: Coiled-coiled templation for live-cell imaging. (**A**) Coiled-coil templation allows for site specific conjugation of an oligonucleotide to a membrane protein via native chemical ligation (**B**).

**Figure 7 biomolecules-12-01523-f007:**
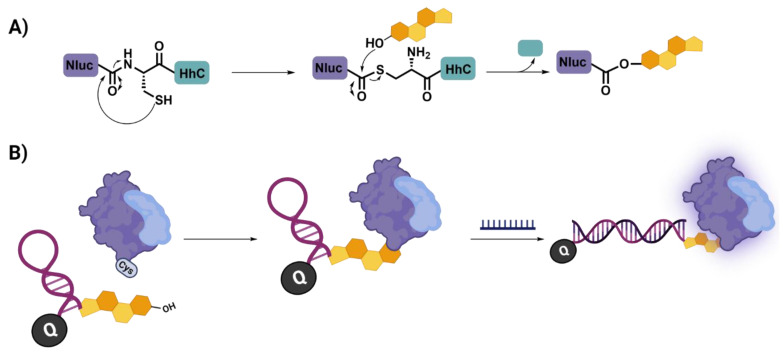
Case Study 4: hijacking post-translational processing to facilitate conjugation. (**A**) the HhC domain undergoes self-splicing and is liberated from an N-terminal fusion protein by the addition of an incoming sterol. (**B**) This can be exploited for the synthesis of nucleic acid sensing enzymatic beacons.

**Figure 8 biomolecules-12-01523-f008:**
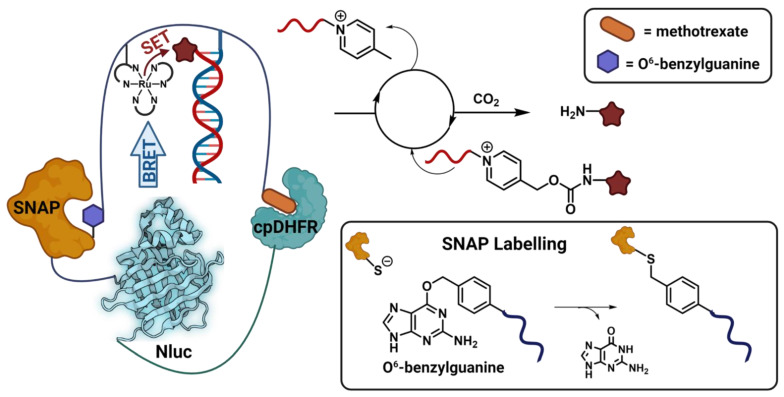
Case Study 5: conjugation facilitates the formation of a precise three-dimensional reaction center for precise drug release. SNAP labelling and ligand templation allow the nLuc luciferase to be brought into close proximity to the photocatalyst to enable BRET-based uncaging of cargo molecules.

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
