# Peer review of "Synthesis of Protein-Oligonucleotide Conjugates"

_biomolecules, 2022, doi:10.3390/biom12101523_

Round 1
Reviewer 1 Report
The authors in the present manuscript offer the reader an overview of the synthesis of protein-oligonucleotide conjugates. This review covers the five complex systems for preparing site-specific chemical conjugation between DNA oligonucleotides and proteins. In my opinion, the manuscript could be slightly improve (for example, there is a lot of abbreviations and some of them are, DBCO, pAzF, not explained in the text) to make it much easier for the readers to follow.
Remarks: Line 86 - UAA pAzF is a mistake pAzF or it should be pAcF? Full meaning of this abbreviation could be written.
Some mistakes with numbering of Figures in the text: Line 175 - should be Figure 4, Line 199 - Figure 5, Line 220 - Figure 6, Line 243 - Figure 7, Line 265 - Figure 8.
Quality of Figures could be better (Figure 6).
In general, this review after minor revision could be interesting for a specific audience on this topic, and I recommend it for publication on MDPI Biomolecules.
Author Response
Line 86 - UAA pAzF is a mistake pAzF or it should be pAcF? Full meaning of this abbreviation could be written. - pAzF is correct and the abbreviation has been added. The abbreviation for DBCO has also been added.
Some mistakes with numbering of Figures in the text: Line 175 - should be Figure 4, Line 199 - Figure 5, Line 220 - Figure 6, Line 243 - Figure 7, Line 265 - Figure 8. – we thank the reviewer for pointing out these errors which are now corrected.
Quality of Figures could be better (Figure 6). – A higher resolution version of figure 6 has been included.
Reviewer 2 Report
This review provides valuable information regarding the most recent strategies for specific conjugation of oligonucleotids to proteins applying chemical and biochemical methods. A case study section helps the reader to gain insight into specific applications.
Some corrections and suggestions are listed:
line 45: "IS (instead of IN) inherently challenging"
line 53 THAT is repeated
line 59 "While naturally occurring moieties present practical ease in terms of obtaining a substrate" may benefit from rephasing
line 62 "challenges in access to the starting protein." may benefit from rephasing
line 86 please define the abbreviation pAzF
lines 105-106 "The promiscuity of these enzymes means that oligonucleotides functionalized with the appropriate tag can be conjugated to the target protein." This sentence is quite misleading. Please rephrase to explain that the enzyme recognises two specific sequences that - if localised on different partners - widen the possible targets.
line 109 "and transfers a tag to it which contains an N-terminal poly-glycine tag". Sortase A is used to connect two molecules (i.e. a protein and an oligonucleotide) if each bears one of the two sequences (LPXTG tag and N-terminal poly-glycine tag) recognised by the enzyme. Rephrasing would help to make the concept clearer.
line 117 "Other classes OF enzymeS used for protein oligonucleotide"
line 132 "efficiently effect conjugation" I suggest the use of AFFECT instead of EFFECT
line 136 "a templation motif" is probably a term not familiar to the general reader
lines 130-145 the strategy described in this paragraph needs explaination
line 171 please correct "The is achieved"
line 175 Figure 3 is Figure 4 in the manuscript
line 182 please provide explaination for the abbreviation DBCO
line 190-191: panels A-C are not explained in the figure legend and not mentioned in text.
199 reference to figure 5 instead of 4
line 212: in figure 5 BPA is represented as a red circle. it could be useful for the reader to be specified in the figure legend.
line 220 reference to figure 6 instead of 5
line 239: panels A and B of figure 6 are not explained in the figure legend and not mentioned in text. a more detailed figure legend would help the reader to understand the steps of this approach.
line 243 reference to figure 7 instead of 6
line 259 panels A and B of figure 7 are not explained in the figure legend and not mentioned in text.
line 244: the term "immolysis" is not familiar to the general reader
line 265 reference to figure 8 instead of 7
line 281: is enableD
Author Response
line 45: "IS (instead of IN) inherently challenging" – corrected
line 53 THAT is repeated - the superfluous THAT has been removed.
line 59 "While naturally occurring moieties present practical ease in terms of obtaining a substrate" may benefit from rephasing – rephrased to “While naturally occurring moieties present practical ease in terms of substrate access”
line 62 "challenges in access to the starting protein." may benefit from rephasing – rephrased to “at the expense of challenges in obtaining the starting protein”
line 86 please define the abbreviation pAzF – the definition has been added
lines 105-106 "The promiscuity of these enzymes means that oligonucleotides functionalized with the appropriate tag can be conjugated to the target protein." This sentence is quite misleading. Please rephrase to explain that the enzyme recognises two specific sequences that - if localised on different partners - widen the possible targets. – We thank the reviewer for pointing this out. The sentence has been rephrased to remove all ambiguity. It is the promiscuity relative to what is appended to the substrate that makes these enzymes interesting.
line 109 "and transfers a tag to it which contains an N-terminal poly-glycine tag". Sortase A is used to connect two molecules (i.e. a protein and an oligonucleotide) if each bears one of the two sequences (LPXTG tag and N-terminal poly-glycine tag) recognised by the enzyme. Rephrasing would help to make the concept clearer. – rephrased to “which recognizes a LPXTG tag on the C-terminus of a target protein and conjugates it to molecules bearing an N-terminal poly-glycine tag”
line 117 "Other classes OF enzymeS used for protein oligonucleotide" – corrected
line 132 "efficiently effect conjugation" I suggest the use of AFFECT instead of EFFECT – this is not correct. This has been corrected using “achieve” instead of effect.
line 136 "a templation motif" is probably a term not familiar to the general reader – changed to templating element, and defined as ‘supramolecular interactions’ in the sentence
lines 130-145 the strategy described in this paragraph needs explanation – the following section was changed to more clearly explain the concept. “This strategy relies on the recognition element (red) which binds to the target and displays an oligonucleotide strand. The complimentary conjugation oligonucleotide strand (blue) then hybridizes with the first strand, bringing it close to the desired region on the protein surface in order to undergo covalent conjugation, followed by dissociation of the templating moiety.”
line 171 please correct "The is achieved" – correct to “this is achieved”
line 175 Figure 3 is Figure 4 in the manuscript – corrected
line 182 please provide explanation for the abbreviation DBCO – the definition has been added
line 190-191: panels A-C are not explained in the figure legend and not mentioned in text. – panels A-C are now explained in the figure legend
199 reference to figure 5 instead of 4 – corrected
line 212: in figure 5 BPA is represented as a red circle. it could be useful for the reader to be specified in the figure legend. – added to the figure legend
line 220 reference to figure 6 instead of 5- corrected
line 239: panels A and B of figure 6 are not explained in the figure legend and not mentioned in text. a more detailed figure legend would help the reader to understand the steps of this approach. - panels A and B are now explained in the figure legend, which has been expanded
line 243 reference to figure 7 instead of 6 – corrected
line 259 panels A and B of figure 7 are not explained in the figure legend and not mentioned in text. - panels A and B are now explained in the figure legend
line 244: the term "immolysis" is not familiar to the general reader – changed to cleavage
line 265 reference to figure 8 instead of 7 – corrected
line 281: is enabled – corrected